# ORN: Inferring patient-specific dysregulation status of pathway modules in cancer with OR-gate Network

**Lifan Liang**[1], **Kunju Zhu**[2], **Junyan Tao**[3], **Songjian Lu**[1¤]*

**1** Department of Biomedical Informatics, University of Pittsburgh, Pittsburgh, Pennsylvania, United States of America, **2** Clinical Medicine Research Institute, Jinan University, Guangzhou, Guangdong, China, **3** Department of Pathology, University of Pittsburgh, Pittsburgh, Pennsylvania, United States of America

¤ Current address: Department of Biomedical Informatics, University of Pittsburgh, Pittsburgh, Pennsylvania, United States of America

* songjian@pitt.edu

**Data Availability Statement:** Processed data from METABRIC can be downloaded at: https://www. cbioportal.org/study/summary?id=brca_metabric Processed data of TCGA-LLG can be downloaded at: https://xenabrowser.net/datapages/?cohort=

## Abstract

Pathway level understanding of cancer plays a key role in precision oncology. However, the current amount of high-throughput data cannot support the elucidation of full pathway topology. In this study, instead of directly learning the pathway network, we adapted the probabilistic OR gate to model the modular structure of pathways and regulon. The resulting model, OR-gate Network (ORN), can simultaneously infer pathway modules of somatic alterations, patient-specific pathway dysregulation status, and downstream regulon. In a trained ORN, the differentially expressed genes (DEGs) in each tumour can be explained by somatic mutations perturbing a pathway module. Furthermore, the ORN handles one of the most important properties of pathway perturbation in tumours, the mutual exclusivity. We have applied the ORN to lower-grade glioma (LGG) samples and liver hepatocellular carcinoma (LIHC) samples in TCGA and breast cancer samples from METABRIC. Both datasets have shown abnormal pathway activities related to immune response and cell cycles. In LGG samples, ORN identified pathway modules closely related to glioma development and revealed two pathways closely related to patient survival. We had similar results with LIHC samples. Additional results from the METABRIC datasets showed that ORN could characterize critical mechanisms of cancer and connect them to less studied somatic mutations (e.g., BAP1, MIR604, MICAL3, and telomere activities), which may generate novel hypothesis for targeted therapy.

## Author summary

Cellular functions are carried out by a set of gene products. Mutation of a single gene is often sufficient to disrupt certain biological functions and promote tumorigenesis. Therefore, genes participating in the same function are less likely to mutate in the same sample. Such phenomenon is called "mutual exclusivity". In this study, our algorithm (ORN) has utilized this property to identify gene-level mutations that affect similar biological

TCGA%20Lower%20Grade%20Glioma%20(LGG)
Processed data of TCGA-LIHC can be downloaded at: https://xenabrowser.net/datapages/?cohort=GDC%20TCGA%20Liver%20Cancer%20(LIHC)
Python implementation of ORN is available at: https://github.com/LifanLiang/ORN

**Funding:** SL is funded by the National Institutes of Health (nih.gov) [R00LM011673, R01LM012011]; the UPMC Hillman Cancer Center Developmental Funding that are supported in part by the National Cancer Institute (cancer.gov) [P30CA047904]. The funders have no role in study design, data collection and analysis, decision to publish, or preparation of the manuscript.

**Competing interests:** The authors have declared that no competing interests exist.

functions. It also considers mutations' impact on mRNA expression. Functional modules identified by ORN tends to be mutually exclusive while causing similar differential expression profiles. When we applied ORN to lower-grade glioma and liver cancer datasets, we have identified gene modules significantly related to patient survival. Furthermore, across different types of cancer, ORN has connected well-known cancer driver mutations with genes whose functions remain unclear. These connections, once validated, can generate novel hypothesis for biologist to further investigate cancer mechanism and develop targeted therapy.

This is a *PLOS Computational Biology* Methods paper.

## Introduction

In the past decades, the emergence of high-throughput technology has dramatically facilitated cancer research. One important insight from these efforts is that although somatic genomic alterations (SGA) vary from patient to patient, they exhibit similar patterns on pathway level [1]. Different somatic mutations in different samples perturb the same signalling pathway and drive similar tumour phenotype. Based on this insight, many anti-cancer therapies targeting specific pathways have been successfully deployed [2]. Unfortunately, most of these therapies are only effective for specific subpopulations. Furthermore, patient responsiveness is often difficult to predict. For example, despite the effectiveness of trastuzumab in HER2 positive patients, some might suffer a relapse for unknown reasons [3].

The current pathway-level understanding of cancer, though valuable, needs further investigation. Many studies [4–8] have only used differentially expressed genes (DEGs) in tumours to analyze dysregulated pathways, either through enrichment analysis or network analysis. However, this approach only looks at pathways affected by gene expression. Pathways leading to such differential expression were usually ignored.

To investigate pathways driving the transcriptomic phenotype, it is natural to consider reconstructing signal transduction among SGA events as biologically plausible (Fig 1A). However, this is extremely difficult due to the dimensionality of high-throughput data. Suppose the conditional independence test is adopted to infer the causal order of SGAs as in the widely used PC algorithm [9]. The required sample size increases dramatically as the number of involved SGAs grows larger [10]. Research efforts dealing with this issue can be categorized into two approaches.

The first approach relies on existing knowledge of signalling networks. Several methods have integrated somatic alteration profiles with prior knowledge to infer pathway activities for individual cancer samples [11–14]. PARADIGM [11] and psSubpathway [13] integrate known pathways as gene networks and inferred pathway status from the observed multi-omics profiles. Junwei et al. [14] utilize protein-protein interaction to reconstruct altered signalling pathways. However, this approach may reduce the chance of obtaining novel biological insights, as our current knowledge is incomplete and biased towards well-studied pathways and genes [15]. Furthermore, a recent study [16] has shown that databases of biological pathways are different from each other. Using different databases yielded disparate results in pathway analysis.

The second approach focuses on an easier task. Instead of the exact signalling order, it searches for groups of SGAs participating in the same pathway, defined as pathway modules in

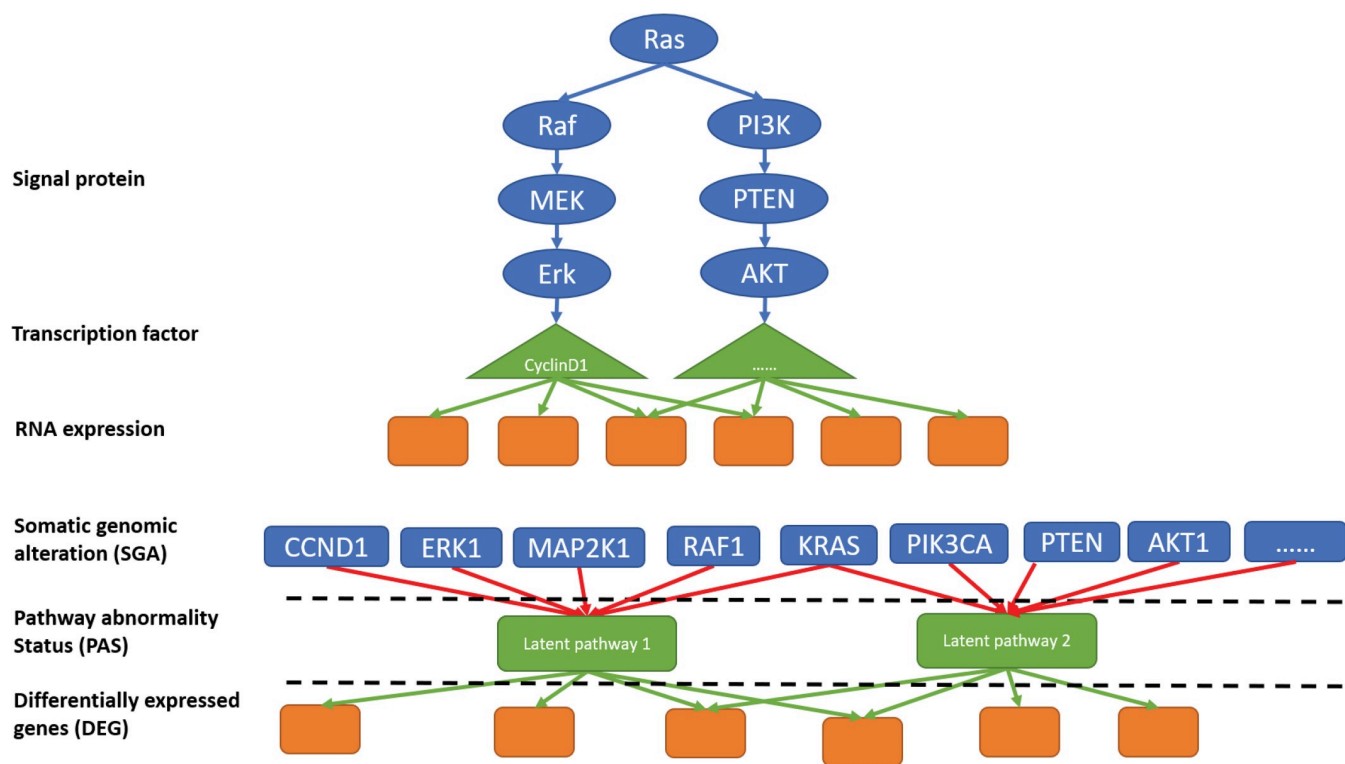

**Fig 1. Illustration of pathway representation by ORN.** Fig 1A is the biologically plausible representation of a signalling network. An SGA event in one of the gene products can disrupt the normal signal cascade. Given the difficulty in estimating the order of signal transduction, we replaced the realistic representation by connecting all the possible SGA events of genes to an OR gate indicating the pathway abnormality status (PAS). After parameter estimation and causal relation extraction, edges with larger weights remain, resulting in Fig 1B. Gene-level SGAs connecting to the same PAS produce the functional module. PAS is then connected to transcriptomics with the same logical OR relationships. Instead of signal transduction or transcription regulation, ORN edges are more abstract, representing noisy logical induction.

this study. Assuming that SGAs in the same pathway drive similar phenotypes, it is possible to identify pathway modules by integrating SGA profiles and corresponding high-throughput phenotypes. Examples of this approach are *de novo* methods [17–19] to identify latent pathways (factors) with multi-omics profiles. To ensure interpretability and computational feasibility, these methods assumed linear relationships among different types of omics data. Thus, it may not be sufficient to capture complicated biological relationships. On the other hand, some de novo methods were directly adopted from the machine learning community [20–22], which may miss some biological features of cellular signalling pathways. For example, mutual exclusivity is a well-known phenomenon of somatic alterations in tumours, which is difficult to be captured by any machine learning model assuming additive effects. In fact, most methods [23–25] model mutual exclusivity with statistical tests.

In this study, we present the OR-gate Network (ORN), a *de novo* model to infer the pathway modules and their corresponding status in cancer. Following the second approach, ORN represented biological pathways (Fig 1A) as a group of SGAs affecting the same pathway abnormality status (PAS) (Fig 1B). In ORN, all SGAs connect to a PAS through a logical OR gate. It indicates that a single SGA is sufficient to dysregulate a pathway (PAS = true) if the inferred causal relation is high. This important property of the ORN agrees with the mutual exclusivity perturbation patterns observed in cancer genomics (i.e., each tumour usually has at most one mutation to perturb a pathway).

Similarly, all PAS connect to a DEG through a logical OR-gate. It indicates that a single pathway is sufficient to cause differential expression of its regulon. A more interesting perspective is viewing ORN in the patient-specific dimension. Each patient has a different combination of SGAs [26]. They exhibit various combinations of abnormal pathways, leading to heterogeneous differential expression patterns. Our previous work [27] showed that the OR-gate mechanism was suitable to capture cancer heterogeneity in gene expression data that a plethora of bi-clustering algorithms pursued [28].

Since the generative process of ORN is consistent with both mutual exclusivity patterns in somatic alteration profiles and sample heterogeneity in transcriptomic profiles, we hypothesize that ORN can recover: (1) gene expressions regulated by the same pathway; (2) groups of SGAs that perturb the same pathway; (3) which pathway is dysregulated in which set of patients. To our knowledge, ORN is the first integrative model capable of inferring patient-specific pathway status while soliciting corresponding SGAs and their regulon.

## Materials and methods

### Overview

Our model contains three layers of variables: somatic genomic alterations (SGA), pathway abnormality status (PAS), and differentially expressed genes (DEG). The probabilistic OR-gate function connects these three layers. As illustrated in Fig 2, the workflow of ORN consists of:

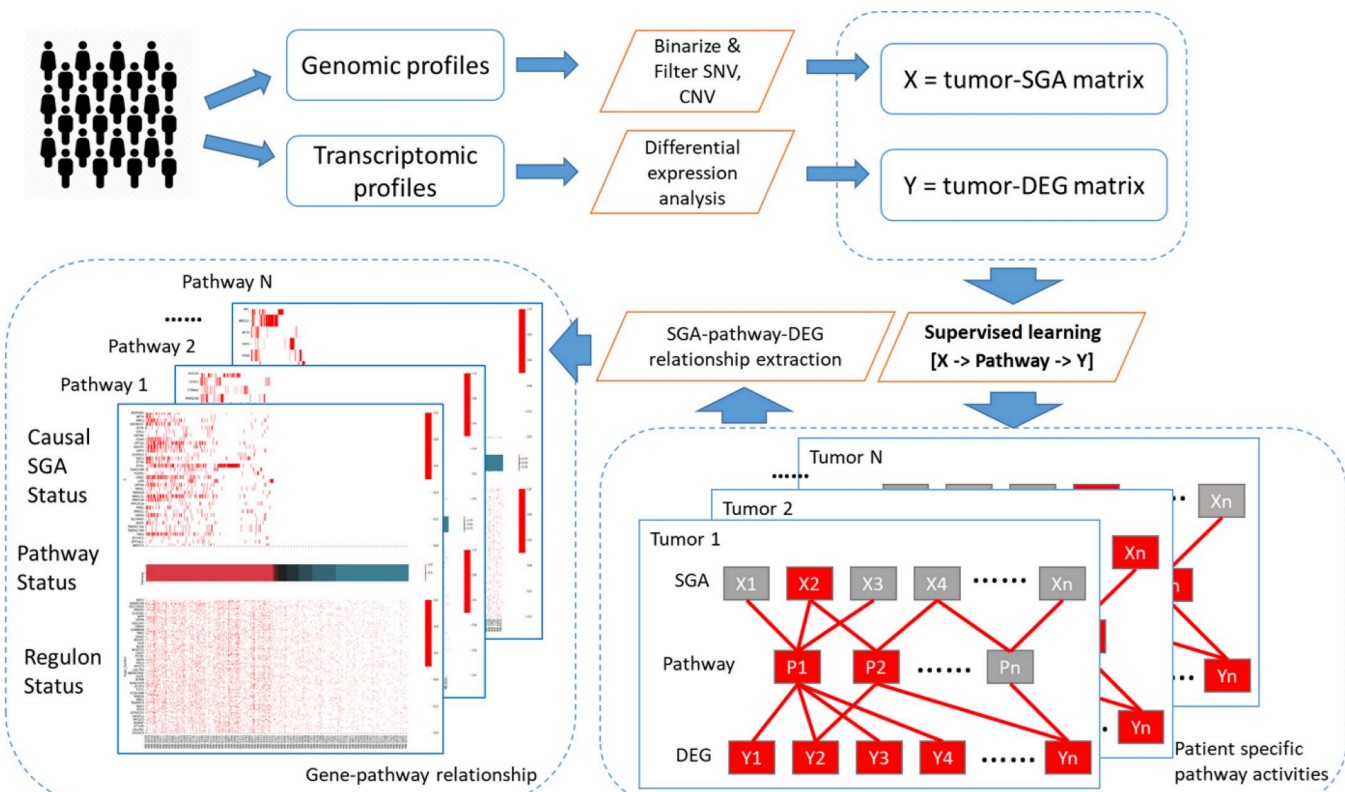

**Fig 2. Workflow overview of ORN.** The input for ORN consists of quantified matrices of single nucleotide variation (SNV), copy number variation (CNV), and gene expression (RNAseq). SNV and CNV were combined, binarized, and filtered on the genes' side to produce a binary event matrix. As for RNAseq, we calculate robust Z score for each gene in each sample. We assumed Logical OR relations when binary events led to pathway dysregulation and, in turn, led to differential expression. ORN algorithm aimed to infer: (1) the relationship between somatic mutations and signalling pathways; (2) the relationship between signalling pathways and differential expressions. With the ORN output, we can recover pathways that were perturbed by somatic mutations and caused differential expression.

(1) preprocessing genomic profiles and transcriptomic profiles of tumour; (2) Inferring causal relations from SGA to PAS to DEG; (3) extracting groups of SGA and regulon related to the same pathway. Each step will be described in detail in the sections below.

## Data preprocessing

The input and output of a probabilistic OR-gate function are required to be Boolean variables. Therefore, before applying ORN to real-world datasets, we need to transform genomic profiles and transcriptomic profiles into binary matrices.

For the somatic alteration profiles, binary values dictate whether a gene has somatically mutated within a sample. We used two types of data: (1) non-silent gene-level single nucleotide variation (SNV) dataset; (2) gene-level copy number variation (CNV). An element in the CNV matrix was set to 1 if its original value was 2/-2. The cutoff for CNV was decided because a looser cutoff, such as 1/-1, does not have strong correlations with gene expression. We then combined CNV and SNV data into a binary event matrix, that is, if the alteration of gene $i$ was observed in either SNV or CNV in sample $j$, then the $ij$th element in the binary event matrix was set to 1.

For the transcriptomic profiles, a binary value dictates whether a gene has differentially expressed within a sample. We first removed genes with median expression counts lower than 10 across all samples to avoid insufficient statistical power. Then the Z scores provided by the CBioPortal [29] platform were binarized. A gene has differentially expressed if its Z score exceeds the range of P-value 0.05. More specifically, an element in the Z score matrix was set to 1 if its absolute value was greater than 1.96, otherwise 0.

To further filter the genes in the SGA level, we applied Multitask Lasso implemented in Scikit-learn [30]. The genomic profiles were used as independent variables and the status of differential expression as targets. Somatic mutations with nonzero coefficients were retained as the input for ORN.

## Probabilistic OR-gate

Before describing the full model, we need to illustrate the key operation unit, probabilistic OR-gate. When there are multiple possible causes $X_1, X_2, \ldots, X_n$ ($X_i \in \{0,1\}$) of an effect variable y (y$\in$ {0,1}), the OR-gate function outputs the probability of y = 1 given the status of $X_1, X_2, \ldots, X_n$, namely:

$$\Pr(y = 1 | X_1, X_2, \ldots, X_n) = OR([X_1, X_2, \ldots, X_n], [X_1 \to y, X_2 \to y, \ldots, X_n \to y])$$

As described by Oniśko et al. [31], OR-gate imposes two assumptions about X and y: (1) each of the causes $X_i$ has a probability Pr ($X_i \to y$) of being sufficient to produce the effect in the absence of all other causes, and (2) the ability of each cause being sufficient is independent of the presence of other causes.

Instead of the binary variable in the original work, this study has generalized $X_1, X_2, \ldots, X_n$ to continuous variables within [0,1]. In this way, $X_i$ represents the probability that the $i$th cause takes place. Note that this notation is equivalent with the binary ones when $X_1, X_2, \ldots, X_n$ are observed. This is the case when we model the impact of SGA on PAS. However, when we infer PAS's impact on DEG, PAS are latent causes that cannot be directly observed from data. Therefore, it is necessary to generalize [$X_1, X_2, \ldots, X_n$] as a vector of probabilities.

For convenience, we will denote the vectorized OR gate function as $OR(\boldsymbol{X}, \delta)$ in all the following content, where $\boldsymbol{X}$ represents the vector [$X_1, X_2, \ldots, X_n$], and $\delta$ represents the vector [Pr

$(X_1 \rightarrow y)$, $\text{Pr}(X_2 \rightarrow y),\ldots,\text{Pr}(X_n \rightarrow y)]$. In this study, we define the OR-gate function as:

$$OR(\boldsymbol{X}, \delta) = 1 - \prod\nolimits_{i=1}^{n}(1 - X_i\delta_i)$$

where $\delta_i$ $(\text{Pr}(X_n \rightarrow y))$ represents the probability that $X_i = 1$ causes $y = 1$. A major difference between the probabilistic OR-gate and the noisy OR-gate [31] is the estimation of $\boldsymbol{X}$ and $\delta$. Previously, $\delta_i$ was estimated by experts or special cases where $X_i = 1$ and all other variables are zero. In this study, $\delta$ is directly learned from data. So is $\boldsymbol{X}$ when PAS is the cause. Their estimation enables us to estimate this integrative model from SGA to PAS to DEG.

Moreover, there are situations where $\boldsymbol{X}$ cannot fully capture the cause of $y$. For example, pathway perturbation in cancer cells may be caused by methylation changes, not just somatic mutations. To deal with unmodeled causes, we followed the original work [31] to introduce the leaky parameter, which is similar to the intercept in linear regression:

$$P_0 = \text{Pr}(X_0 = 1)\text{Pr}(X_0 \rightarrow y)$$

where $X_0$ is the unmodeled cause, assumed to stay active across different samples (i.e. $\text{Pr}(X_0 = 1) = 1$). Causal relations between $X_0$ and $y$, $\text{Pr}(X_0 \rightarrow y)$, still needs to be learned in the model. Hence the probabilistic OR gate function becomes:

$$\text{Pr}(Y = 1 | X_1, X_2, \ldots, X_n) = 1 - [1 - \text{Pr}(X_0 \rightarrow Y)] \prod\nolimits_{i=1}^{n}[1 - \text{Pr}(X_i = 1)\text{Pr}(X_i \rightarrow Y)]$$

During implementation, $P_0$ can be modelled by adding a row into the binary event matrix with all ones, which represents that $Pr(X0 = 1) = 1$.

## OR-gate network

With the probabilistic OR-gate function, we connected SGA to PAS and then to DEG, as illustrated in Fig 1B. Let us denote S as the number of samples, M the number of genes in genomics, P the number of pathways, and G the number of genes in transcriptomics. Let Mut be the S×M binary event matrix of somatic mutation, Path be the S×P matrix representing PAS for each tumour sample, and Expr the S×G RNA expression matrix. Let U denote the M×P causal relationship matrix between SGA and pathways. Let Z denote the P×G causal relationship matrix between pathways and DEG. Note that Path dictates the binary status of the pathway module (dysregulated or not).

For the $s$th sample, the activity of the $p$th pathway is:

$$Path_{sp} = OR(Mut_{s.}, U_{.p})$$

where $Mut_{s.}$ is the $s$th row of the matrix Mut and $U_{.p}$ is the $p$th column of the matrix U). This formulation implied that all SGAs are connected to multiple OR-gate functions; then each OR-gate determined one PAS. The matrix U implied the causal relationship between SGAs and PAS. In other words, the set of genes in a certain pathway can be identified from U. Therefore, the OR gate is similar to the "collider" shape in a Bayesian network. When the common effects (PAS) are learned, mutual exclusivity among SGAs will naturally occur.

And the status of the $g$th DEG in the $s$th sample is:

$$\widehat{Expr}_{sg} = OR(Path_{s.}, Z_{.g})$$

In this formulation, DEG was caused by PAS through the OR-gate function. OR-gate functions adopted in this layer is mainly to model sample heterogeneity in gene expression dataset. The binary matrix Z implied the causal relations between patient-specific PAS and DEGs.

## Gradient-based parameter estimation

Given the relationship matrix U and Z, the aggregate pathway status can be computed with the OR gate function. Therefore, we only need to estimate U and Z in the model. These two types of parameters are estimated by maximizing the likelihood of observed gene expression given the generative process of ORN. That is, the objective function to optimize for ORN is the overall log likelihood of the observed Expr given estimated Expr:

$$LL(Expr) = \sum_{s \le S, g \le G} [Expr_{sg} \log \widehat{Expr}_{sg} + (1 - Expr_{sg}) \log(1 - \widehat{Expr}_{sg})]$$

where $\widehat{Expr}_{sg}$ is the probability of differential expression of the $g$th gene in the $s$th sample computed by ORN. Similar latent variable models, such as LDA [32], are usually computationally expensive with MCMC or variational inference. However, we found that the layered structure of ORN is similar with the neural network architect. Thus, the learning algorithm essential to all deep learning models, back propagation, can be used to estimate ORN parameters.

First, we need to reparameterize U and Z such that the parameters are not bounded within [0,1]. That is, every element in the matrix U and Z is regarded as an output of a scalar within $[-\infty, +\infty]$:

$$U_{mp} = sigmoid(\mu_{mp})$$

$$Z_{pg} = sigmoid(\zeta_{pg})$$

where $\mu$ and $\zeta$ are matrices with the same shape as U and Z, but their values are unconstrained. This enables us to apply gradient-based methods to identify maximum likelihood of ORN regarding $\mu$ and $\zeta$.

During implementation, we adopted the Rprop algorithm [33] to learn $\mu$ and $\zeta$. This algorithm requires the gradient of $\zeta$ with respect to $LL(Expr)$:

$$\frac{\partial LL}{\partial \zeta_{pg}} = Z_{pg}\left(1 - Z_{pg}\right) \sum_{n \le N} \left[ \frac{Path_{sp}\left(1 - \frac{Expr_{sg}}{\widehat{Expr}_{sg}}\right)}{1 - Path_{sp}Z_{pg}} \right]$$

Using the chain rule, we can also derive the gradient of $\mu$ with respect to $LL(Expr)$:

$$\frac{\partial LL}{\partial \mu_{mp}} = \sum_{s \le S} \frac{\partial LL}{\partial Path_{sp}} \cdot \frac{\partial Path_{sp}}{\partial U_{mp}} = \sum_{s \le S} \frac{\partial LL}{\partial Path_{sp}} \cdot \frac{Mut_{sm}U_{mp}(1 - U_{mp})(1 - Path_{sp})}{1 - Mut_{sm}U_{mp}}$$

where the computation of $\partial LL/\partial Path_{sp}$ is symmetric to $\partial LL/\partial Z_{sp}$.

To control the sparsity of parameters, we assume U and Z are samples from the Beta distribution. Thus, the gradients above need to be modified. For example, the partial derivative of $\zeta$ should be modified as:

$$\frac{\partial LL}{\partial \zeta_{pg}^*} = \frac{\partial LL}{\partial \zeta_{pg}} + (\alpha - 1)(1 - \zeta) + (\beta - 1)\zeta$$

where $\alpha$ and $\beta$ are the hyperparameters for Beta distribution. For all the experiments in this study, we set $\beta = \alpha = 0.95$. The procedure for model estimation has been summarized in Fig 3.

```
Algorithm 1: ORN inference
   Input  : Mut, a S × M binary matrix; Expr, a S × G binary matrix; P
            number of latent pathways; α, β, Beta priors; MaxIter,
            maximum iterations for the gradient descent
   Output: U, a M × P binary matrix; Z, a P × G binary matrix; Path, a
            S × P binary matrix
 1  μ^{M×P} ← Gaussian(mean = 0, std = 0.1);
 2  ζ^{P×G} ← Gaussian(mean = 0, std = 0.1);
 3  i ← 0;
 4  while i < MaxIter do
 5  │   U ← sigmoid(μ);
 6  │   Z ← sigmoid(ζ);
 7  │   Path ← OR(Mut, U);
 8  │   X̂ ← OR(Path, Z);
 9  │   G_μ^{M×L}, G_ζ^{N×L} ← ComputeGradient(Mut, Expr, U, Z, α, β);
10  │   μ, ζ ← RPROP(μ, ζ, G_μ, G_ζ);
11  │   i ← i + 1;
12  end
13  U ← sigmoid(μ);
14  Z ← sigmoid(ζ);
15  Path ← OR(Mut, U);
16  return U, Z, Path
```

**Fig 3. The pseudo code to compute the two relationship matrices U and Z, and the pathway activities.**

### Causal relation extraction

After learning the model parameters, $\mu$ and $\zeta$, we can recover U and Z through the element-wise sigmoid function of $\mu$ and $\zeta$. The matrix of patient-specific PAS, *Path*, can be recovered by computing the OR gate function given SGAs and U.

When applied to real-world datasets, we also need to extract pathway modules and regulons corresponding a latent pathway. The pathway module was determined by the matrix U. If $U_{mp}>0.5$, then we concluded that mutation of gene $m$ could disrupt pathway $p$. From an operational perspective, a pathway module corresponding to pathway $p'$ would be genes with $U_{mp'}>0.5$.

Regulon related to a latent pathway is extracted similarly. If $Z_{pg}>0.1$, then we conclude that the disruption of pathway p can cause differential expression of gene g. In this way, the set of genes regulated by the same pathway are grouped into a coexpression module. Note that for

real data analysis in this study, the cutoff for elements in Z was the top 5% value among all genes in the pathway $p$.

Please note that for convenience, the module of SGAs and the modules of DEGs corresponding to a latent pathway will simply be called "upstream module" and "downstream module" respectively in following sections.

Since there is no ground truth for real data analysis, we performed Gene Ontology (GO) enrichment analysis on the downstream modules to characterize the functional impacts of the upstream modules.

## Simulation and evaluation

Synthetic data was generated by following the probabilistic OR gate mechanism. First, somatic mutations and the two relationship matrices were generated with Bernoulli distribution. Then *Path* and *DEG* were generated by performing noisy OR-gate computation with $P_0 = 0$. To simulate the mutual exclusivity patterns observed in real data, we performed post pruning. When several mutations belonging to the same pathway took place in the same sample, all but one of them were removed.

The artificial neural network (NN) was used as a baseline to evaluate ORN's efficacy of inferring pathway activities. To ensure the neural network model was comparable with ORN, it had one hidden layer, and the activation function was sigmoid. In this way, the values of hidden neurons were also within [0,1]. We ran NN and ORN on 20 synthetic datasets and computed their reconstruction error and the Jaccard score of pathways. Reconstruction error was computed as:

$$\text{error} = \frac{\sum_{s \leq S, g \leq G} |\widehat{Expr_{sg}} - Expr_{sg}|}{S \times G}$$

We further propose Jaccard score to evaluate how ORN's performance changes in various settings. Unlike reconstruction error, this criterion measures the similarity between the inferred relationship matrix and the ground truth. To compute Jaccard score, we first need to compute Jaccard similarity for U and Z respectively:

$$sim(U_{.p}^*) = max_{p' \leq P}[Jaccard(U_{.p}^*, U_{.p'})]$$

$$sim(Z_{p.}^*) = max_{p' \leq P}[Jaccard(Z_{p.}^*, Z_{p'.})]$$

where $U^*$ and $Z^*$ are the true relationship matrix in synthetic data. The function Jaccard(A, B) takes the form:

$$Jaccard(A, B) = A \cdot B / (A + B - A \cdot B)$$

Then the Jaccard score for one dataset is:

$$Jaccard\ score = \sum_{p \leq P} \frac{sim(U_{.p}^*) sim(Z_{p.}^*)}{P}$$

## Public datasets

We performed ORN on the high-throughput data of 514 lower-grade glioma patients. It was downloaded from The Cancer Genome Atlas (TCGA)[34]. After preprocessing, 511 samples remained with 15501 DEGs and 670 SGAs. The number of pathway modules was set to 10.

To evaluate how different cancers may have perturbed the same pathways, we also applied ORN to the METABRIC dataset [35] downloaded from CBioPortal. It contained 2173 samples

of somatic variant profiles and 1904 samples of RNA-seq. After preprocessing, we retained a binary event matrix with 1092 genes and a binary DEG matrix with 24256 genes. Both matrices had 1866 samples. ORN was trained with 15 pathway modules.

We also tested whether ORN can identify pathways related to patient survival in liver cancer samples. Both somatic alteration profiles and transcriptomic profiles were downloaded from The Cancer Genome Atlas (TCGA) [34]. After preprocessing, 506 samples remained with 17779 DEGs and 607 SGAs. The number of pathway modules was set to 15.

## Results

### ORN was effective in recovering OR-gate relationships

Synthetic data were generated according to the generative process described in "Simulation and evaluation". The number of pathways was set to 5; The number of samples, SGAs, and DEGs were all set to 1000. This was referred to as the standard setting.

We proposed Jaccard score and reconstruction error to evaluate the performance. Jaccard score can measure the concordance between inferred relationship matrices and the ground truth. Details of calculation was described in "Simulation and evaluation".

From the standard setting, each condition was changed separately to see how they affected the performance. As shown in Fig 4, ORN has achieved almost perfect recovery (>99%) in the standard setting. However, ORN's performance dropped over 20% when the number of samples dropped from 1000 to 500, or the number of mutations increased from 1000 to 3000 mutations. Note that when the number of DEGs were reduced to 500, the performance remained the same. Interestingly, the performance of ORN also decreased and became unstable when more pathway modules were needed.

### ORN provided more insights than the neural network

We cannot identify similar algorithms that only used high-throughput data to infer pathway activities. However, we found that artificial neural networks (NN) with sigmoid activation function can also produce binary values in the hidden layer. In addition, both ORN and NN used backward propagation to optimize parameters. Thus, we designed a neural network architecture similar with ORN and used it as a baseline.

In the synthetic experiment, although NN converged to comparable reconstruction error as ORN, its accuracy was in pathway recovery was only around 50% (Fig 5). This showed that NN is less capable of capturing all the signals in the data.

Similar results were observed when we applied NN to the glioma dataset (described in the next section). The relationship matrix estimated with NN is much more redundant than ORN. GO enrichment analysis of the downstream modules (see S1 Table) also showed that NN could only capture less than 5 major aspects with 10 hidden neurons, while ORN can cover different biological aspects of glioma with each pathway module. As shown in Fig 6, the relationship matrix learned by NN contains much redundancy (Fig 6A), while each pathway in ORN regulated different sets of genes with few overlaps (Fig 6B). This indicated that the biological mechanism from somatic mutations to transcriptomic profiles could be more accurately characterized by the OR-gate logic imposed by ORN rather than conventional non-linear relationships.

### ORN detected pathways closely related to patient survival

After applying ORN to the lower-grade glioma dataset, survival analysis showed that pathway 6 and pathway 7 had significant impacts on patient survival (Fig 7). We performed Gene Ontology (GO) enrichment analysis on the top DEGs in these pathways (see S2 Table). The

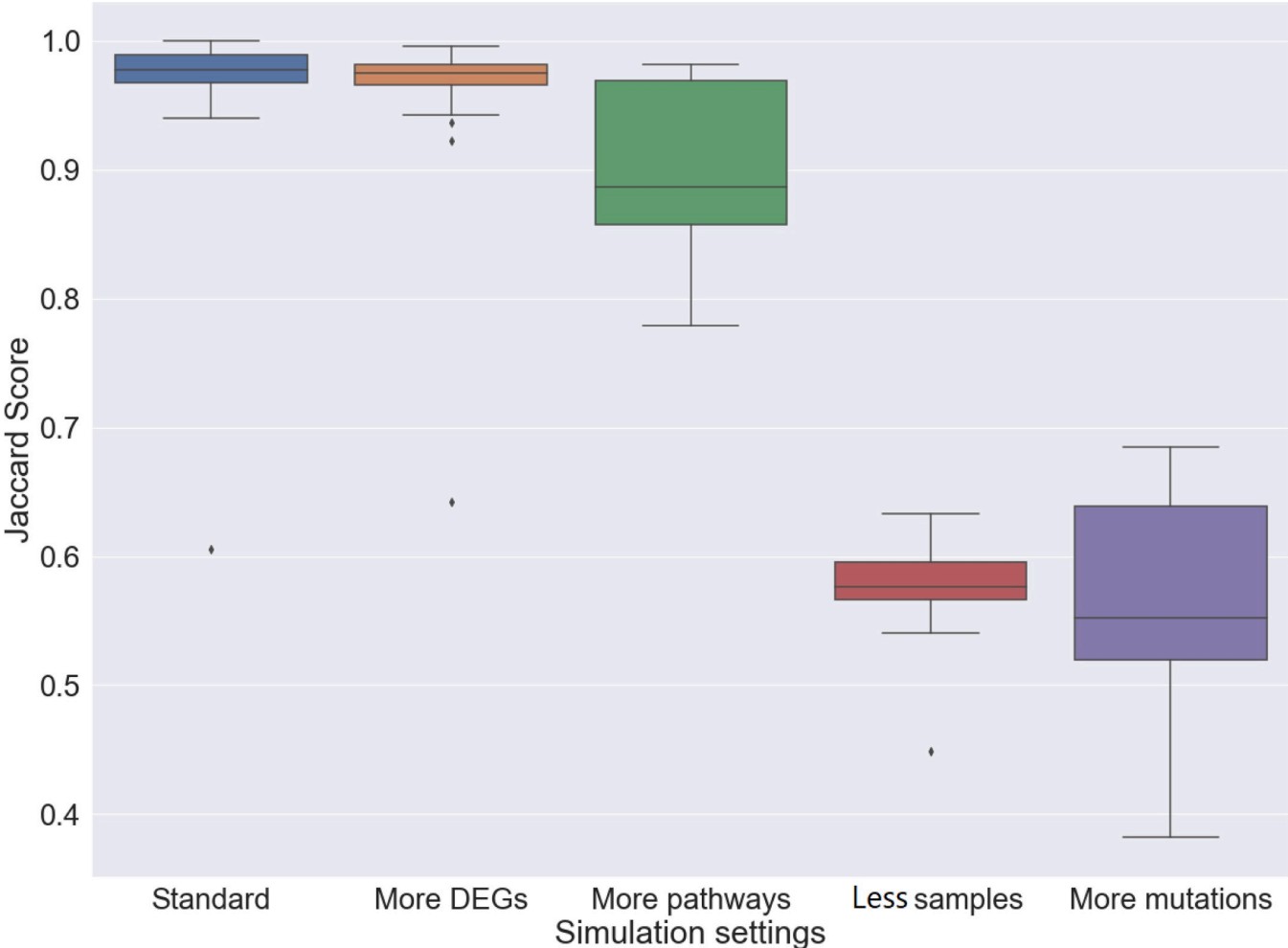

**Fig 4. Performance of ORN across different settings.** In the standard setting, ORN has recovered pathway modules with almost perfect accuracy. Reducing the number of DEGs did not affect the performance of ORN. Adding more pathway modules would introduce more variation to ORN's performance. When the number of samples decreased to 500 or the number of mutations increase to 3000, the median Jaccard score has decreased to 73% and 81% respectively.

downstream module of pathway 6 is mostly related to DNA processing activities. Cancer samples with this pathway dysregulated probably have compromised genomic instability [36], leading to worse survival. Its upstream module includes CDK13 [37], H3F3A [38], IDH1 [39], PTEN [40], SNRPE [41] that are closely related to DNA repair or DNA replication.

As for pathway 7, we found that PTEN, H3F3A, and POM121L12 were shared by the upstream modules in both pathways. However, the top 300 DEGs caused by the two pathways have no genes in common. GO enrichment analysis showed that downstream modules are related to neutrophil activities, Ras signal transduction, and viral genome replication. We conjectured that cancer samples with pathway 7 dysregulated exhibited viral infection and its immune response. Since virus infection can drive glioma formation [42], This subgroup of patients may be more likely to progress to malignancy and worse survival.

When applying ORN to liver cancer samples, we also identified two pathways related to patient survival (Fig 8). One is pathway 2. GO enrichment analysis of the upstream alterations showed that pathway 2 mainly affected epithelial tube formation (GO:0072175) and other activities located on the membrane (cytoskeletal anchoring and protein localization). Epithelial

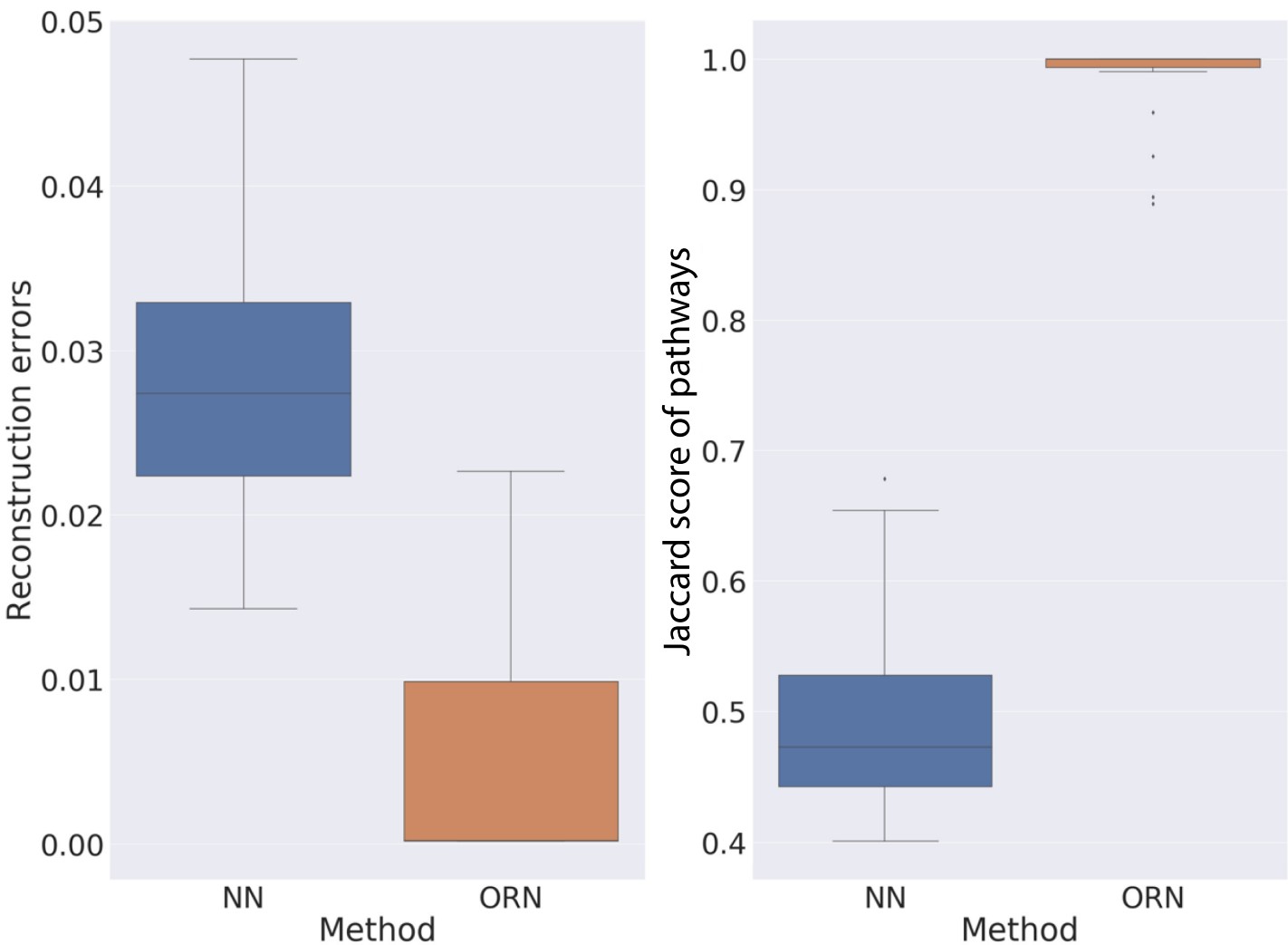

**Fig 5. Comparison of ORN and NN on synthetic datasets.** Boxplot on the left showed the distribution of prediction error of NN and ORN across 20 synthetic experiments. Boxplot on the right showed the distribution of cosine similarity between the inferred pathways and ground truth.

tube formation contributed to epithelial to mesenchymal transition (EMT) and mesenchymal to epithelial transition (MET) that played a vital role in liver cancer development and metastasis [43]. the overlapped gene Podocalyxin (PODXL) was found to be overexpressed in HCC cell line and could be used as a biomarker to predict the prognosis of HCC due to participating in HCC migration and invasion processes [44]. Fibrosis growth factor receptor 2 (FGFR2) (also in neuron projection morphogenesis, GO:0048812) and its partner driver genes are frequently found in intrahepatic cholangiocarcinoma (ICC) [45,46]. This pathway module was also enriched for the regulation of cardiac conduction (GO:1903779). One gene related to cardiac conduction, ASPH, is highly overexpressed in cholangiocarcinoma(CCA) and HCC [47]. Inhibition of ASPH could decrease CCA development [48]. Another related gene, ITPR2, is the major intracellular calcium release channel in hepatocytes to regulate Calcium (Ca2+) signaling, resulting in regulating lots of function of hepatocytes, including glucose and lipid metabolism, apoptosis, gene transcription, bile secretion, and cell proliferation [49]. ITPR2 was also found to be decreased in the fatty liver with impaired liver regeneration [49].

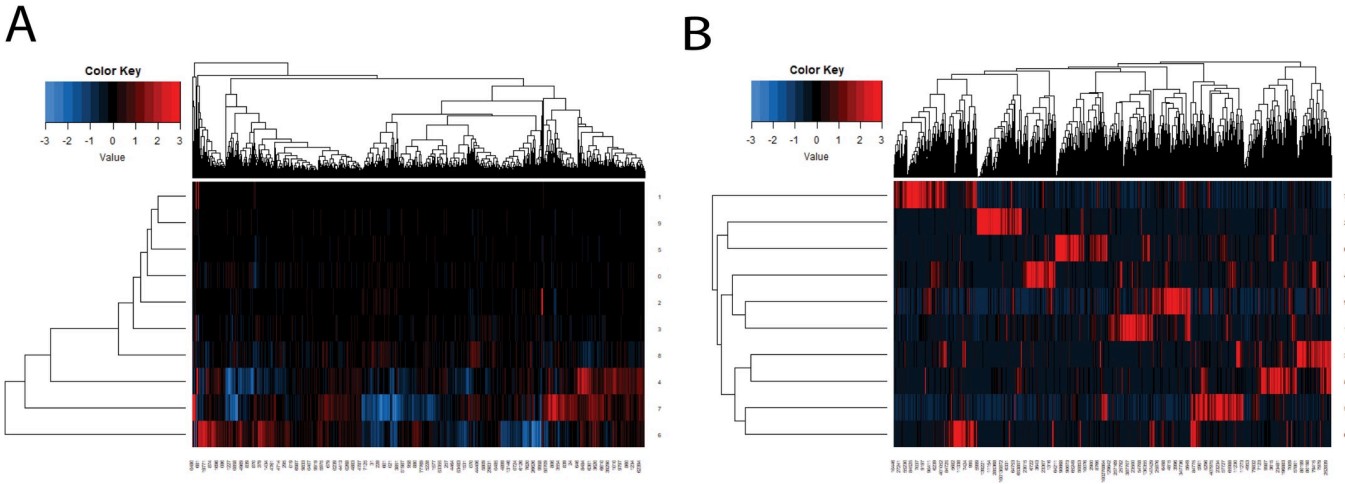

**Fig 6. Heatmap representation of the relationship matrix between pathways and differential expression after row normalization.** The relationship matrix generated by a neural network (NN) contains many redundant signals, while ORN automatically pushes for sparsity. Each pathway module in ORN has uniquely caused a subset of genes to express differentially.

The other one, pathway 3, is related to progress free interval. Enrichment analysis shows that both upstream mutations and downstream regulons participate in the lectin pathway. Few studies have investigated the association between lectin and liver disease. A recent study [50] shows that expression c type lectin plays an important role in different stages of chronic liver disease. It is possible that lectin is important for the immune response within tumour environment. Several genes related to lectin pathway were shown to be important in liver cancer. For example, KRAS is usually found mutated in CCA patients [51,52]. The proto-oncogene

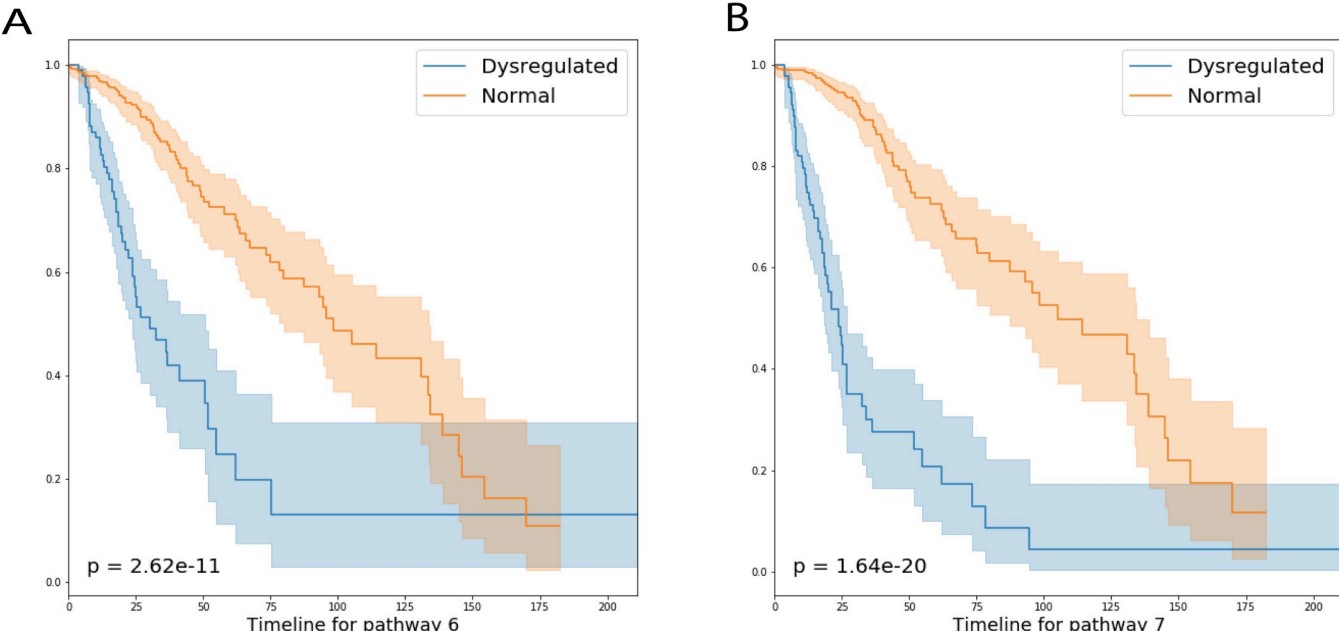

**Fig 7. Survival analysis of LGG patients with and without pathway dysregulation.** LGG patients with pathway 6 (A) and pathway 7 (B) dysregulated have worse overall survival than those without. X-axis is in the unit of month; Y-axis represents the proportion of each subgroup. 102 patients' pathway 6 were dysregulated, 100 patients' pathway 7 were dysregulated. Both groups have 62 samples in common.

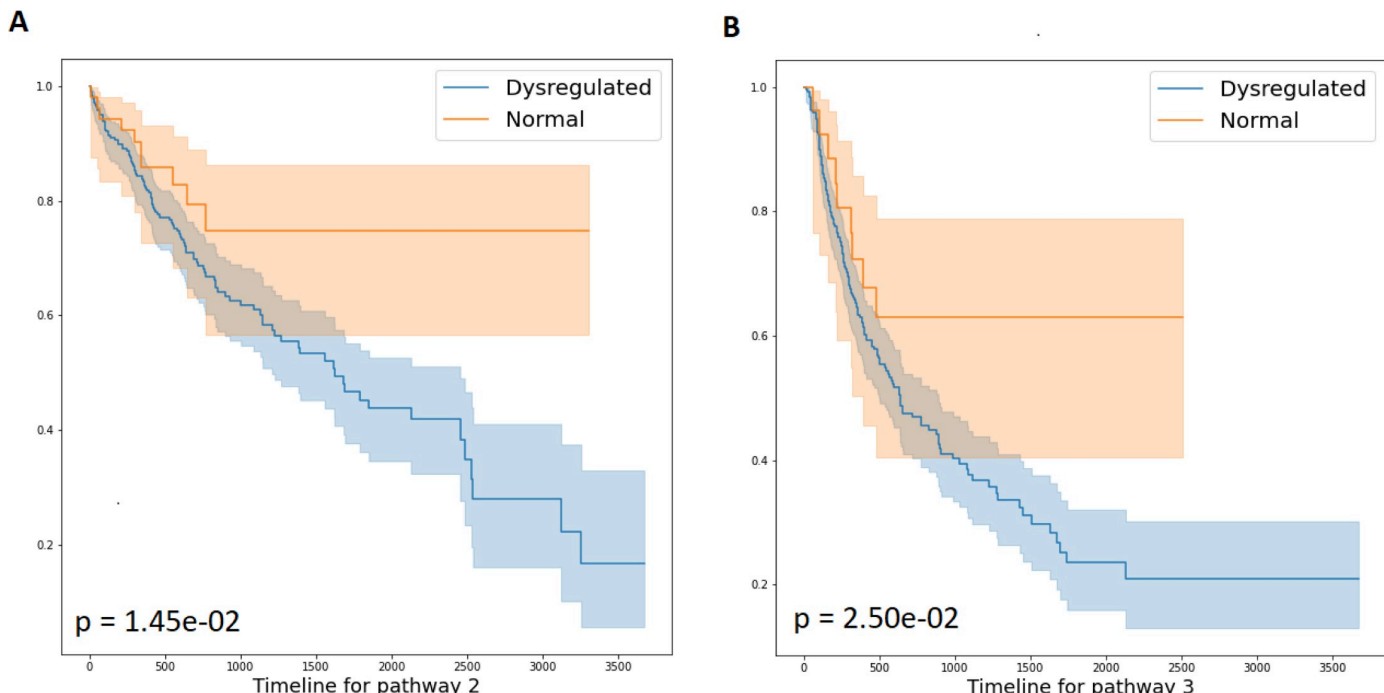

**Fig 8. Survival analysis of liver cancer patients with and without pathway dysregulation.** Liver cancer patients with pathway 2 (A) dysregulated have worse overall survival than those without. Dysregulation of pathway 3 also results in worse progression-free interval.

tyrosine kinase Src is usually aberrantly expressed in HCC with an effect on cell proliferation, differentiation [53–56]. Meanwhile, mucin 5AC (MUC5AC) as a secreted Mucins was upregulated in ICC and CHC patient and inflammation [57]. It is a good diagnostic marker in CCA and a good biomarker to differentiate CCA from benign biliary diseases [58,59]. The upstream module in pathway 3 is also enriched for cation transport (GO:0006812). Genes related to cation transport were also important for liver cancer. For example, LRP2 is involved in fusion in HCC patient [60], and some research found complement C3 concentration changes occurred at the very early stage of tumourigenesis in serum proteins of diethylnitrosamine (DEN). 2-AAF was found to induce Wistar rats tumour model[61]. It might be a novel therapeutic approach for liver cancer [62].

## ORN characterized common mechanisms in cancer

As for the METABRIC dataset results, upstream modules of six pathway modules were almost identical and hence merged into one union (S3 Table). This module contained well-known oncogenes such as TP53, PTEN, PIK3CA, and MAP3K1. The corresponding pathway was dysregulated across all samples (probability above 0.5). This pathway likely represents the common cause of breast cancer. GO enrichment analysis (see S3 Table) showed that its corresponding downstream module was mostly involved in mitosis, such as G1/S transition of mitotic cell cycle phase transition (GO:0044772). Pathway 5 also exhibits similar downstream effects but had different SGAs. Most notably, MIR604 was in the upstream modules of pathways. Studies has shown that polymorphism of MIR604 is related to the development of hepatocellular carcinoma [63] and the metastasis of colorectal cancer [64]. MIR604 is differentially expressed in breast cancer [65]. Still, to our knowledge, the impact of MIR604 mutation in breast cancer has not been investigated. In the case of LIHC, both the upstream and downstream modules in pathway module 1 were significantly enriched with nucleotide-excision

repair (GO:0006289) and DNA repair (GO:0006281). This pathway contained TP53BP1, ERCC3, BRCA2, which are well known for DNA maintenance activities.

In the glioma dataset, we found pathway 4 to be closely related to immune response. Downstream modules of pathway 4 were enriched for various immune responses, including cytokine-mediated signalling and toll signalling. This subgroup of patients may not be responsive to immune therapy. Within this pathway module is the mutation of interferon alpha 21 (IFNA21), which plays an important role in inflammatory responses and toll signalling. IFNs are also identified as major factors of patient response to various cancer therapies [66]. Moreover, we found that PTK6 and SRMS within the same upstream module. The product of two genes work closely together as intracellular kinases [67] and promotes invasive prostate cancer [68]. However, they are rarely studied in the context of glioma and immune response.

Like glioma, one particular pathway in breast cancer captured abnormal immune response in a subgroup of cancer samples. The downstream module in pathway 3 was related to immune response, including T cell activation (GO:0042110), regulation of immune response (GO:00507006), inflammatory response (GO:0006954). The upstream module included CDC20, COLEC12, MED8, MPL, SOX5, and OTUD1. CDC20 is known to be related to T cell activation. COLEC12's protein product is associated with innate immunity [69]. SOX5 is shown to be related to B cell proliferation [70]. Another interesting gene is MED8. Studies show that MED8 is important to regulate resistance against bacteria in plants [71]. Meanwhile, MED8 is implicated in renal cell carcinoma. However, it is rarely investigated in the case of breast cancer and innate immunity. As for OTUD1, a recent study [72] has shown that its induction by RNA virus may inhibit the innate immune response.

## ORN detected pathway dysregulation specific to cancer types

Although not related to patient survival, other pathways in glioma samples also captured different aspects of molecular characteristics. For example, pathway 0 is closely related to the biosynthesis of cholesterol, steroid, and alcohol, while cholesterol metabolism has recently been studied as a potential therapeutic target [73]. Besides, downstream modules of pathway 0 contained differentially expressed genes enriched for central nervous system development. In the corresponding upstream module, we identified SZT2 [74] and TIAM1 [75] to be closely related to nervous system development. Other mutations, such as CPAMD8 and RUBP1, exhibited mutual exclusivity and similar expression patterns. Yet, these two genes have not been studied in terms of central nervous system development.

As for the breast cancer samples, the pathway module 8 contained several well-known driver mutations, including KRAS, APC, and ARID1A. As shown in Fig 9, most genes in this module exhibited mutual exclusivity, while these mutations caused differential expression of a similar set of genes. The coexpression module was enriched for telomere and t-circle formation, which were well-known factors for cancer initiation and tumour survival [76]. In the pathway module, the relation between telomere and APC [77], KRAS [78], ARID1A [79], PRKG1 [80] has been reported. Although mutually exclusive to the four genes above, we have not found research linking BAP1, MIR604, and MICAL3 to telomere activities. Visualization of all other pathway modules are included in S1–S40 Figs.

## Discussion

In this study, we proposed ORN, the first de novo method to infer patient-specific pathway activities from genomic profiles to transcriptomic profiles of cancer patients. Compared with the traditional neural network method, ORN provided much more insights into how the somatic mutations function together. In a traditional neural network, the activation of a node

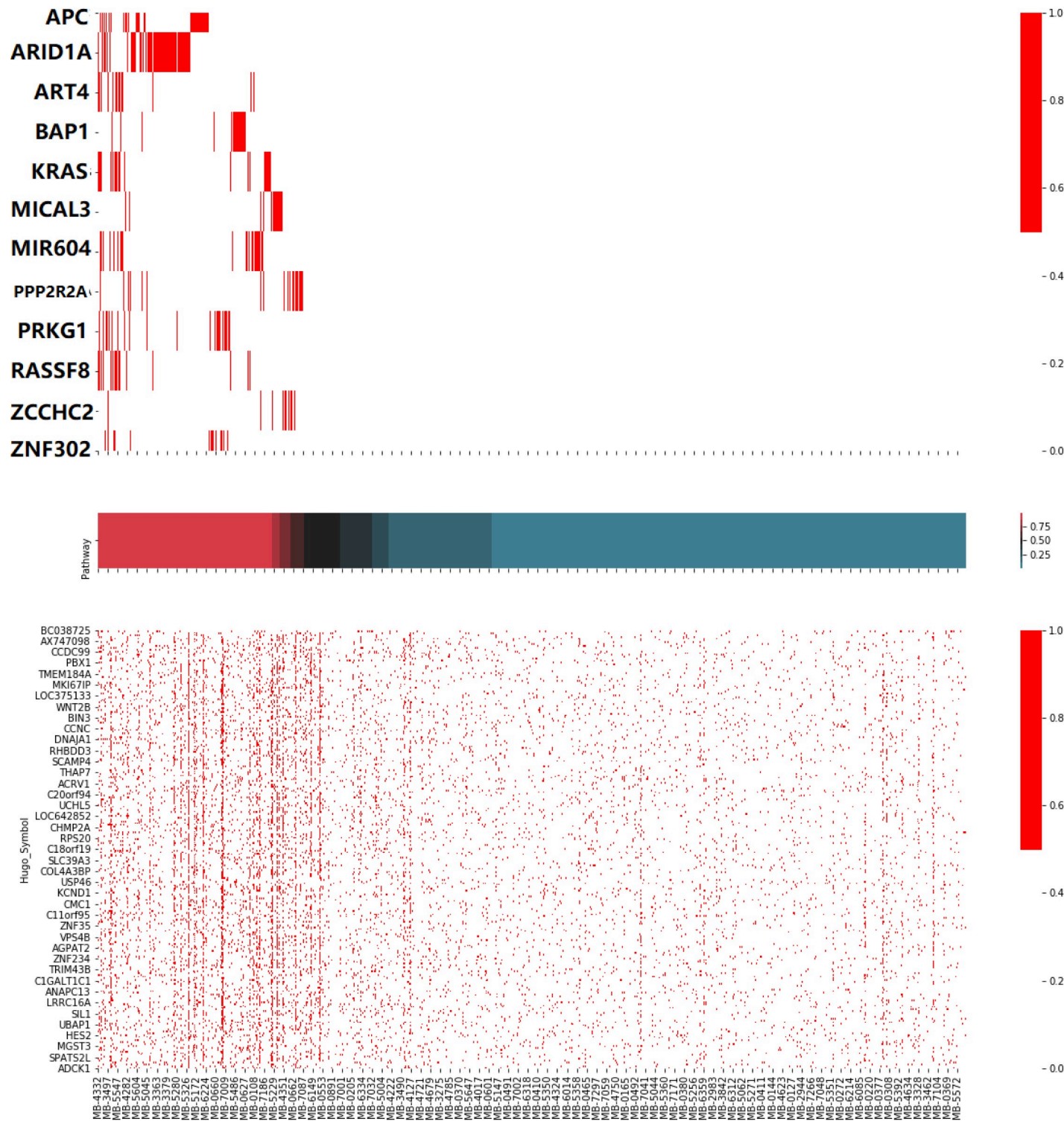

**Fig 9. Illustration of pathway 8 in breast cancer samples.** Cancer samples sorted by PAS (middle figure) was the X-axis shared by the three subplots. The upper figure showed the mutation event of the upstream module (cutoff 0.5), while the bottom figure showed the heatmap of differential expression of the downstream module (roughly top 1% related). The top figure showed patterns of mutual exclusivity, while the bottom showed a strong correlation between pathway activities and differential expression.

is decided by the accumulation of all inputs. In contrast, the output of a node in ORN would be true if any input to this node is true. Hence, the ORN agrees with the premise of mutual exclusivity of somatic alteration in tumours.

Meanwhile, OR-gate also allows the co-occurrence of genes within the same pathway. This flexibility enables the model to handle the rare situation where genes within the same pathway mutated together. Still, when the pathway modules were known, the relations between SGA and pathways established the "collider" shape well known in a Bayesian network. Thus, when learned with backpropagation, ORN tends to identify the mutual exclusivity patterns.

Besides mutual exclusivity, there are many other patterns or biological mechanisms in cancer biology. For example, if two mutations co-occurred in most samples, then they are likely to disrupt two different pathways causing tumour. This co-occurrence pattern is best captured by AND-gate instead of OR-gate. However, the output of AND-gate should be the occurrence/ progression of tumour. Such information is difficult to be integrated into our model. In the future, we may consider modifying the model to integrate other relevant information.

Please note that the "pathway module" formulated in this study may not exactly correspond to a known pathway. In the real data analysis, we showed that a pathway module might represent the impact of somatic alterations on immune response and microenvironment (i.e. pathway 4 in glioma and pathway 3 in breast cancer). In addition, a pathway module may also encapsulate the joint status of multiple pathways. For example, pathway 0, 1, 2, 7, 9, and 14 in breast cancer probably captured a common set of dysregulated pathways. That is because ORN is performed on the sample level. If several biological pathways were dysregulated on the same sets of patients, ORN cannot distinguish them.

In the real data analysis with lower-grade glioma and breast cancer, ORN has recovered major mechanisms consistent with current knowledge, such as abnormal DNA repair ability and immune response. Glioma patients with these dysregulated pathways had lower survival rates. ORN further revealed mechanisms specific to cancer types, such as the steroid metabolism and nervous system development in glioma. We identified several somatic mutations that might be related to certain malfunctions in cancer cells, worthy of further biological investigation. However, ORN requires in-depth analysis to obtain useful insights. In the future, we will try to develop statistical tests to automatically return meaningful genes in both upstream and downstream modules.

For the METABRIC dataset, none of the PAS inferred by ORN is significantly related to patient survival. We conjectured that the reason was two-fold: (1) Somatic mutations may not be the only source of variation of RNA expression. Sharma et al. [81] showed that copy number alterations, epigenetic changes, transcription factors, and microRNAs collectively explain, on average, only 31–38% and 18–26% expression variation; (2) compared with glioma samples, the cellular constitution in breast cancer samples was probably more diverse. To handle the first issue, we need to include more data sources in a principled way. In the future, epigenetic profiles may be included to inform the coregulation of RNA expression. As for the second issue, future research needs to incorporate reliable complete deconvolution algorithms in the data preprocessing step.

For some genes in the pathway module, we failed to find evidence supporting their functional associations to the affected DEGs. Although some of them were likely to provide novel molecular insights, many were false positive. Upon closer investigation, we believed there are two major sources of false positives: (1) passenger mutations that exclusively occur in highly mutated samples. For example, ACSS1 was in most pathway modules of glioma because it only occurred in highly mutated samples, which had most pathways dysregulated. (2) passenger mutations within the same copy number variation event as driver mutations. When a set of genes mutated in almost the same set of patients, it is likely that only one of them contributed to the pathway dysregulation.

During analysis, we also found that different pathway modules may share a subset of somatic mutations. For example, pathway 6 and pathway 7 in glioma have PTEN in common. Thus, it is possible that hierarchical structures of SGA functions can be inferred from the overlapping pathway modules. In the future, we may provide more convenient visualization utilities to analyze the hierarchies among pathway modules. In addition, so far, we have not found an effective measure to identify the appropriate number of pathway modules. We encourage future research on the balance between model complexity and model likelihood of ORN.

We proposed ORN to infer pathway modules and their abnormality status from high-throughput profiles of cancer samples. Application of ORN in lower-grade glioma, breast cancer, and liver cancer detected pathway modules closely related to patient survival. ORN also connected somatic mutations to key mechanisms of cancer, such as DNA repair and innate immune response. Although some mutations' function (e.g., MIR604) was not supported by literature, they were mutually exclusive to well-known driver mutations and caused differential expression in a similar subset of genes. We encouraged biological researchers to use ORN to infer personalized pathway activities and generate novel hypotheses for targeted therapy.

## Supporting information

**S1 Fig. Illustration of LLG pathway 0.**
(EPS)

**S2 Fig. Illustration of LLG pathway 1.**
(EPS)

**S3 Fig. Illustration of LLG pathway 2.**
(EPS)

**S4 Fig. Illustration of LLG pathway 3.**
(EPS)

**S5 Fig. Illustration of LLG pathway 4.**
(EPS)

**S6 Fig. Illustration of LLG pathway 5.**
(EPS)

**S7 Fig. Illustration of LLG pathway 6.**
(EPS)

**S8 Fig. Illustration of LLG pathway 7.**
(EPS)

**S9 Fig. Illustration of LLG pathway 8.**
(EPS)

**S10 Fig. Illustration of LLG pathway 9.**
(EPS)

**S11 Fig. Illustration of LIHC pathway 0.**
(EPS)

**S12 Fig. Illustration of LIHC pathway 1.**
(EPS)

**S13 Fig. Illustration of LIHC pathway 2.**
(EPS)

**S14 Fig. Illustration of LIHC pathway 3.**
(EPS)

**S15 Fig. Illustration of LIHC pathway 4.**
(EPS)

**S16 Fig. Illustration of LIHC pathway 5.**
(EPS)

**S17 Fig. Illustration of LIHC pathway 6.**
(EPS)

**S18 Fig. Illustration of LIHC pathway 7.**
(EPS)

**S19 Fig. Illustration of LIHC pathway 8.**
(EPS)

**S20 Fig. Illustration of LIHC pathway 9.**
(EPS)

**S21 Fig. Illustration of LIHC pathway 10.**
(EPS)

**S22 Fig. Illustration of LIHC pathway 11.**
(EPS)

**S23 Fig. Illustration of LIHC pathway 12.**
(EPS)

**S24 Fig. Illustration of LIHC pathway 13.**
(EPS)

**S25 Fig. Illustration of LIHC pathway 14.**
(EPS)

**S26 Fig. Illustration of METABRIC pathway 0.**
(EPS)

**S27 Fig. Illustration of METABRIC pathway 1.**
(EPS)

**S28 Fig. Illustration of METABRIC pathway 2.**
(EPS)

**S29 Fig. Illustration of METABRIC pathway 3.**
(EPS)

**S30 Fig. Illustration of METABRIC pathway 4.**
(EPS)

**S31 Fig. Illustration of METABRIC pathway 5.**
(EPS)

**S32 Fig. Illustration of METABRIC pathway 6.**
(EPS)

**S33 Fig. Illustration of METABRIC pathway 7.**
(EPS)

**S34 Fig. Illustration of METABRIC pathway 8.**
(EPS)

**S35 Fig. Illustration of METABRIC pathway 9.**
(EPS)

**S36 Fig. Illustration of METABRIC pathway 10.**
(EPS)

**S37 Fig. Illustration of METABRIC pathway 11.**
(EPS)

**S38 Fig. Illustration of METABRIC pathway 12.**
(EPS)

**S39 Fig. Illustration of METABRIC pathway 13.**
(EPS)

**S40 Fig. Illustration of METABRIC pathway 14.**
(EPS)

**S1 Table. LLG pathway enrichment analysis.**
(XLSX)

**S2 Table. METABRIC pathway enrichment analysis.**
(XLSX)

**S3 Table. LIHC pathway enrichment analysis.**
(XLSX)

## Acknowledgments

We would like to thank Xinghua Lu, Cooper Greg, George Tseng, and Vanathi Gopalakrishnan for stimulating conversions on the topic of causal inference and statistical learning.

## Author Contributions

**Conceptualization:** Songjian Lu.

**Formal analysis:** Junyan Tao.

**Funding acquisition:** Songjian Lu.

**Investigation:** Kunju Zhu, Junyan Tao.

**Methodology:** Lifan Liang.

**Project administration:** Songjian Lu.

**Software:** Lifan Liang.

**Supervision:** Songjian Lu.

**Validation:** Kunju Zhu.

**Visualization:** Lifan Liang.

**Writing – original draft:** Lifan Liang.

**Writing – review & editing:** Songjian Lu.

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
