## [Decision Letter · Decision Letter 0]

3 Sep 2020

Dear Dr. Lu,

Thank you very much for submitting your manuscript "ORN: Extracting latent pathway activities in cancer with OR-gate network" for consideration at PLOS Computational Biology.

As with all papers reviewed by the journal, your manuscript was reviewed by members of the editorial board and by several independent reviewers. In light of the reviews (below this email), we would like to invite the resubmission of a significantly-revised version that takes into account the reviewers' comments.

We cannot make any decision about publication until we have seen the revised manuscript and your response to the reviewers' comments. Your revised manuscript is also likely to be sent to reviewers for further evaluation.

Sincerely,

Tim Beißbarth, Ph.D.

Guest Editor

PLOS Computational Biology

Florian Markowetz

Deputy Editor

PLOS Computational Biology

Reviewer's Responses to Questions

**Comments to the Authors:**

Reviewer #1: The authors Liang et al. describe in the manuscript “ORN: Extracting Latent Pathway Activities in Cancer with OR-gate Network” describe an approach to identify patient specific networks by correlating mutations and gene expression levels.

The manuscript is complete and divided consistently in introduction, methods, results and discussion. It is written in a clear and direct way.

My comments are:

As the authors state and show in Fig. 1 the topology of the network is not represented by an ORN, however, this is the basic characteristic of a network. This does not affect the prediction of the nodes involved in the network, only the naming may not be the best.

The method is designed for the mutual exclusivity perturbation pattern and the test data for the first simulation is pre-pro accordingly. Also, it might be a common pattern, what happens if the situation in the patient differ from this pattern?

If two different pathways are affected by different mutations but in the same patient, is the method able to assign the DEGs to these pathways or are they merged to a set of affected nodes? In the results different pathways are discussed, in the methods it should be explained a bit more

It is understandable to use for the neuronal network the same number of layers. However, different methods may need different number of layers for the best results. How do the results from the NN change if more layers are used?

Reviewer #2: The ORN method of the paper is new; it can be applied to cancer genomic and transcriptomic data, to infer latent pathway activities. The approach can potentially contribute a lot to the field of cancer pathway studies. It will be more convincing to have additional testing datasets of other cancers.

Minor comments:

Page 6, the sentence seems incomplete: "Furthermore, in this novel model with the mechanism from latent pathway y activities to transcriptomics, a pathway perturbed in a subset of patients cause differential expression in a subset of genes."

Reviewer #3: my review is uploaded as an attachment

**Have all data underlying the figures and results presented in the manuscript been provided?**

Reviewer #1: Yes

Reviewer #2: Yes

Reviewer #3: Yes

PLOS authors have the option to publish the peer review history of their article (what does this mean?). If published, this will include your full peer review and any attached files.

Reviewer #1: No

Reviewer #2: No

Reviewer #3: No
---

## [Decision Letter · Decision Letter 1]

7 Dec 2020

Dear Dr. Lu,

Thank you very much for submitting your manuscript "ORN: Extracting Patient Specific Dysregulation Status of Pathway Modules in Cancer with OR-gate network" for consideration at PLOS Computational Biology.

As with all papers reviewed by the journal, your manuscript was reviewed by members of the editorial board and by several independent reviewers. In light of the reviews (below this email), we would like to invite the resubmission of a significantly-revised version that takes into account the reviewers' comments.

We cannot make any decision about publication until we have seen the revised manuscript and your response to the reviewers' comments. Your revised manuscript is also likely to be sent to reviewers for further evaluation.

Sincerely,

Tim Beißbarth, Ph.D.

Guest Editor

PLOS Computational Biology

Florian Markowetz

Deputy Editor

PLOS Computational Biology

Reviewer's Responses to Questions

**Comments to the Authors:**

Reviewer #1: The authors submitted a reworked version of their manuscript titled "ORN: Extracting Patient Specific Dysregulation Status of Pathway Modules in Cancer with OR-gate network". They responded the comments of the reviewers for the first version thoroughly and addressed all questions and remarks.

I have no further comments and recommend a publication.

Reviewer #2: The authors have addressed the comments; They have sufficiently improved their paper.

Reviewer #3: With this revision, the authors slightly improved the manuscript. Regretfully, I still have serious concerns on the form (mainly lack of clarity and rigour of the description of the proposed procedure), which prevent me from making a serious assessment of the content and results. Unfortunately, this revised manuscript contains too many typos, sentences that are poorly worded and problematic use of the past tense.

For example,

1) The definition of a “pathway module” is still lacking and, in my opinion, the concept is still unclear ; does it corresponds to the entries of a gate in the ORN (as suggested in Fig2)? It is however briefly discussed in the discussion, which in my opinion is not the right place to explain it.

2) Figure 1 and related text (paragraphs “In this work, we present OR-gate Network...”, and “Similarly, we use OR-gate to model how pathway...” page 2 including the figure caption) should be seriously revised.

Fig 1A is supposed to represent an example of a signalling cascade (where molecular components are not genes but their products, please correct this in the description), leading to the activation of transcription factors, which in turn controls the transcription of target genes.

Fig1B is misleading as it is. I my opinion, it does not provide support for understanding the ORN model.

a)The inputs of the gate should be drawn differently than the molecular components of the singalling cascades in Fig 1A as they correspond to the genes (whereas in Fig1A, we had their products).

b)I don’t see why the TF would be the gate, rather it should also be an input of the gate (i.e. if CCND1 is mutated, it will affect the expression of its targets...). There should be an OR gate (not corresponding to a molecular component). I suppose that this is the “additional variable indicating the pathway status”?

c)The levels should be named appropriately. What is the genomic level? What is the pathway status (is it given by the output of the gate?)

3) Section Probabilistic OR-gate, page 4

All this section requires a complete and carefully revision. In particular, the notation is not coherent, and there are too many typos: Xi (i should be a subscript), capital vs small letter (e.g. for Y or y), bold letter for vector X (in OR(X,delta) but not in the text), first index of X, 0 or 1 (in “and delta represents the vector [Pr(X0->Y...”)... And in the last formula what X_other represents?

Please check lines 9 in Fig 3 (what are the dimensions L and N?).

4) the equation defining the error needs some explanation. I don’t understand the subscripts sg.

**Have all data underlying the figures and results presented in the manuscript been provided?**

Reviewer #1: Yes

Reviewer #2: Yes

Reviewer #3: Yes

PLOS authors have the option to publish the peer review history of their article (what does this mean?). If published, this will include your full peer review and any attached files.

Reviewer #1: No

Reviewer #2: No

Reviewer #3: No
---

## [Decision Letter · Decision Letter 2]

15 Feb 2021

Dear Dr. Lu,

We are pleased to inform you that your manuscript 'ORN: Inferring Patient-specific Dysregulation Status of Pathway Modules in Cancer with OR-gate Network' has been provisionally accepted for publication in PLOS Computational Biology.

Best regards,

Tim Beißbarth, Ph.D.

Guest Editor

PLOS Computational Biology

Florian Markowetz

Deputy Editor

PLOS Computational Biology

Reviewer's Responses to Questions

**Comments to the Authors:**

Reviewer #1: The authors Liang et al. submitted a new revision of their manuscript “ORN: Inferring Patient-specific Dysregulation Status of Pathway Modules in Cancer with OR-gate Network”. The revision consists mainly of linguistic improvements, the methods and the results are unchanged. The changes improve the language so that the manuscript can be understood without problems. Besides the linguistic changes also the structure and clarity of the text improved by more consistent definitions and naming of the concepts.

The authors addressed and solved the comments of the reviewers; therefore, I don’t see any reason not to publish the manuscript.

**Have all data underlying the figures and results presented in the manuscript been provided?**

Reviewer #1: Yes

PLOS authors have the option to publish the peer review history of their article (what does this mean?). If published, this will include your full peer review and any attached files.

Reviewer #1: No

---

## [Editor Report · Acceptance letter]

22 Mar 2021

PCOMPBIOL-D-20-01283R2 

ORN: Inferring Patient-specific Dysregulation Status of Pathway Modules in Cancer with OR-gate Network

Dear Dr Lu,

I am pleased to inform you that your manuscript has been formally accepted for publication in PLOS Computational Biology. Your manuscript is now with our production department and you will be notified of the publication date in due course.

With kind regards,

Alice Ellingham
